# Advances towards Understanding the Mechanism of Action of the Hsp90 Complex

**DOI:** 10.3390/biom12050600

**Published:** 2022-04-19

**Authors:** Chrisostomos Prodromou, Dennis M. Bjorklund

**Affiliations:** School of Life Sciences, Biochemistry and Biomedicine, John Maynard Smith Building, University of Sussex, Falmer, Brighton BN1 9QG, UK; dennis.bj90@gmail.com

**Keywords:** chaperone, co-chaperone, heat shock proteins, Hsp90, Aha1, immunophilins, p23, Cdc37, kinase, steroid hormone receptor, structure, mechanism, ATPase

## Abstract

Hsp90 (Heat Shock Protein 90) is an ATP (Adenosine triphosphate) molecular chaperone responsible for the activation and maturation of client proteins. The mechanism by which Hsp90 achieves such activation, involving structurally diverse client proteins, has remained enigmatic. However, recent advances using structural techniques, together with advances in biochemical studies, have not only defined the chaperone cycle but have shed light on its mechanism of action. Hsp90 hydrolysis of ATP by each protomer may not be simultaneous and may be dependent on the specific client protein and co-chaperone complex involved. Surprisingly, Hsp90 appears to remodel client proteins, acting as a means by which the structure of the client protein is modified to allow its subsequent refolding to an active state, in the case of kinases, or by making the client protein competent for hormone binding, as in the case of the GR (glucocorticoid receptor). This review looks at selected examples of client proteins, such as CDK4 (cyclin-dependent kinase 4) and GR, which are activated according to the so-called ‘remodelling hypothesis’ for their activation. A detailed description of these activation mechanisms is paramount to understanding how Hsp90-associated diseases develop.

## 1. Introduction

Heat shock protein 90 (Hsp90) is an ATP-dependant chaperone that is subject to regulation by a variety of co-chaperones [1,2]. Hsp90 consists of three domains, a C-terminal domain (CTD) that is inherently dimerised, a middle domain (MD) and an ATP-binding N-terminal domain (NTD) (Figure 1A). ATP binding and hydrolysis drive conformational changes that lead to the NTDs undergoing cycles of dimerisation and disassembly (recently reviewed [3,4,5,6]). During this cycle, the formation of a catalytically active unit leads to the hydrolysis of ATP. The chaperone cycle is thought to be the basis by which Hsp90 client proteins are regulated or matured into an active state, and it is thought that Hsp90 assists in the late stages of the folding of its client proteins [7,8]. The rate-limiting step of the cycle is the co-ordinated structural changes required to bring about the hydrolysis of ATP [9,10]. Client proteins include kinases, such as ErbB2, Cdk4 and Braf, as well as nuclear receptors, transcription factors and structural proteins, such as actin and tubulin. A full list of such protein clients can be found at https://www.picard.ch/downloads/Hsp90interactors.pdf (accessed on 16 April 2022).

The NTD is responsible for the binding of ATP and, together with the catalytic-loop arginine within the MD of Hsp90, they form a catalytically active unit able to hydrolyse ATP [5,11]. The NTD is the target of ATPase regulation by interaction with a series of co-chaperones, including p23 (Sba1 in yeast), Cdc37 and Aha1. The catalytic-loop arginine is essential for ATP hydrolysis and contacts the γ-phosphate of the N-terminally bound ATP [11]. Because of the essential nature of the catalytic-loop arginine, co-chaperones such as Cdc37, p23 and Aha1 may influence Hsp90′s ATPase activity through interactions with the MD of Hsp90 [5,11,12,13]. In addition to these co-chaperones, many bind Hsp90 using a TPR domain that engages the conserved MEEVD motif of Hsp90 at its extreme c-terminus. Such co-chaperones include immunophilins such as FKBP51 (**FK506**-binding protein 51) and FKBP52, HOP (Sti1 in yeast), the phosphatase PP5, the E3 ligase CHIP, AIP and Tah1, amongst others. These co-chaperones may be either specific for a client protein or a class of client or may impart enzymatic activities that regulate Hsp90 for the activation or degradation of a client protein [3,4,14].

The structure and catalytic cycle of the Hsp90 chaperone has been extensively reviewed [2,3,14,15,16], and we direct readers to these. The current manuscript concentrates on advances in understanding the Hsp90 mechanism through recent structural and biochemical studies.

## 2. Concerted and Unconcerted Actions of the ATPase Cycle of Hsp90

The regulation of the ATPase cycle of Hsp90 was recently reviewed [3,4,14]. Formation of catalytically active Hsp90 involves the binding of ATP, which triggers a number of conformational changes. These include closure of the so-called n-terminal lid of each protomer, which traps the bound nucleotide and allows n-terminal dimerisation, an N-terminal β-strand exchange between the two protomers of the Hsp90 dimer, the association of the MD and NTD of each protomer and the release of the catalytic loop from the MD. The association of the N- and middle-domains (N/M-domains) is critically dependent on Arg 380 (yeast Hsp90) [9]. Mutation of Arg 380 to alanine abolishes lid closure, the β-strand swap, and N/M-domain association altogether [9]. Arg 380 interacts with the γ-phosphate of the bound ATP molecule [11] but additionally forms a salt bridge with the catalytic Glu 33 (yeast Hsp90) (Figure 1A). Arg 380, therefore acts as an important interaction site linking together ATP and the catalytic Glu 33 residue. Since Glu 33 is located on helix 2, which is directly involved in the interface between the n/m-domains, it would appear that Arg 380 acts as a sensor detecting the presence of bound ATP and the presence of Glu 33 in the ATP-bound conformation, which subsequently allows N/M association and provides stability to the N/M-domain interface.

Recently, structural changes at local sites within Hsp90 were probed using a reporter system based on fluorescence quenching by photo-induced electron transfer (PET) with nanosecond single-molecule fluorescence fluctuation analysis [9,17]. Unlike fluorescence resonance energy transfer (FRET), which occurs at 2–10 nm scales, PET quenching relies on van der Waals contacts at distances of ≤1 nm, which occur between the organic fluorophore and a tryptophan indole side-chain. The PET reporter system relies on the introduction of a Trp and Cys amino acid residue by site-directed mutagenesis, where the Cys residue acts as a point of fluorophore attachment, and Trp for fluorescence quenching. Three conformational switches were designed: lid-closure using two PET reporter systems (S51C-A110W and A110C-S51W), N/M-domain intra-subunit association (E192C-N298W) and N-terminal β-strand cross-subunit swapping (using a fluorophore on the N-terminus of one subunit (A2C) and Trp on the other (E162W)). The study showed that the ATPase activity of Hsp90 was correlated with the kinetics of the specific structural rearrangements that acted cooperatively on a sub-millisecond time scale. However, lid closure over the nucleotide-binding pocket was identified as a two-step mechanism (Figure 1B). The finding suggests that the lid of apo-Hsp90 is dynamic and populates an ensemble of conformers, which is then reconfigured rapidly, following ATP binding, to an intermediate state that most likely exposes the n-terminal dimerisation interface. Subsequently, the lid fully closes over the nucleotide-binding pocket, which occurs slowly and in a concerted manner with the n- and m-domain inter- and intra-subunit associations. The subunit swap of the n-terminal β-strands upon association of the n-domains, is facilitated by their highly mobile nature, as seen in apo-Hsp90. Consequently, the swapping of the terminal β-strands, the closure of the lid, and n- and m-domain associations are cooperatively coordinated to form the catalytically active Hsp90 conformation that can hydrolyse ATP. Ultimately, the hydrolysis of ATP opens the molecular clamp and reconstitutes the chaperone for its next catalytic cycle (Figure 1B).

The authors also showed that Aha1 substantially accelerated conformational changes. A ~40-fold acceleration in the mean rate constant of n/m-domain association was seen, which was in agreement with the enhanced ATPase activity. In contrast, a ~20-fold increase in the mean rate constant of lid closure and β-strand swapping was observed. Preorganization of the NTD and MD induced by Aha1 could perhaps explain the stronger acceleration of n/m-domain association compared to the other motions [18,19]. The effect of Aha1 was also investigated on the three conformational switches. As previously seen, Aha1 stimulates the ATPase activity of the Hsp90 F349A mutant (F349 is important for N/M association) to wild-type levels [18,19] and could accelerate all three of the local motions. The authors show that binding of Aha1 to apo-Hsp90 showed fluorescence quenching both for the wild-type and mutant F349A protein, suggesting that binding of Aha1 can influence the conformation of the lid. The most recent studies suggest that Aha1 may help “bypass” a slowly formed closed-lid intermediate of Hsp90 [18]. Although other work suggests that Aha1 initiates a partially closed lid conformation and acts late on the ATP-bound n-terminally-dimerised conformation [20]. However, PET-FCS provided evidence that Aha1 could mobilise the lid early in the apo-state of Hsp90 and this is also supported by NMR chemical shift perturbations that show weak and transient interactions of C-Aha1 with the NTD of Hsp90 [21]. The T101I lid mutation can also be re-activated by Aha1 [19], and consequently, it appears that lid mobilisation is an early mode of Aha1 action. Furthermore, it was shown that the mutations T101I and R380A (yeast), which disrupt Hsp90 ATPase activity, also abolish lid closure, β-strand swapping and N/M-domain association altogether, although the lid segment was still able to remodel rapidly. In contrast, the activating A107N mutation accelerated lid closure and N/M-domain association by ~5-fold, in agreement with previous studies [6,11]. Interestingly, the A107N effect on the β-strand swap was moderate.

The PET technique was further developed and combined with two-colour fluorescence microscopy. This allowed the simultaneous detection of two structural coordinates, one colour per coordinate, within a single protein molecule [9]. These experiments showed that the lid of the NTD and the n-terminal β-strand are highly mobile, with μs reconfiguration time constants [22]. These studies concluded that ATP binding rapidly remodels the lid, likely exposing the n-terminal dimerisation interface and priming inter-subunit dimerisation early in the cycle [22]. In conclusion, the recent data suggests a concerted mechanism involving a number of conformational switches that from the catalytically active state of Hsp90 (Figure 1B). However, mutations show that some decoupling of these switches is possible and this is evident with the effect of A107N on the rate of β-strand swapping, which was moderate in comparison to lid and n/m-domain association. Furthermore, Aha1 can accelerate n/m-domain association over that of lid-closure and the β-strand swapping of wild-type Hsp90. Consequently, Hsp90, through a cooperative mechanism, is highly efficient at establishing an active state but nonetheless can be modulated by co-chaperones and perhaps by client proteins themselves to meet the specific demands of client protein activation. Although the dimerisation of Hsp90 is cooperative, the hydrolysis of ATP by each monomer appears to be unaffected by the state of the adjacent protomer. Studies using a variety of Hsp90s show that despite having two ATP-binding sites, their ATPase activities follow simple, non-cooperative kinetics [23,24,25,26]. Consistent with such observations, the activity of the wild-type ATPase domain is unaffected by the adjacent protomer carrying a mutation that is either unable to bind or hydrolyse ATP [27,28]. An adaptation to the above model is seen with the TRAP1 (tumor necrosis factor receptor-associated protein 1) paralogue of Hsp90 [29]. In this model, ATP binding leads to an asymmetric dimer in which one protomer is buckled compared to the other, which remains in a similar conformation to the yeast Hsp90–Sba1-bound structure [11]. In this model the hydrolysis of ATP is sequential and deterministic, where the buckled protomer is better able to hydrolyse ATP. Subsequently, a flip in the MD and CTD asymmetry positions the opposite protomer in the buckled conformation, thus promoting hydrolysis of the second ATP and allowing TRAP1 to proceed through the cycle to its open state.

## 3. The Hsp90–Cdc37–CDK4 Complex

Cdc37 is an Hsp90 co-chaperone that delivers client kinases to Hsp90 and in doing so inhibits the ATPase activity of Hsp90 to allow client protein loading [30]. Structural work has shown how Cdc37 binds to the lid-segment of the NTDs of Hsp90 and prevents them from closing, which in turn prevents n-terminal dimerisation (PDB 1US7) [12] (Figure 2A). In addition, a key interaction between Arg 167 of Cdc37 and the catalytic residue Glu 33 (yeast) of Hsp90 inactivates the catalytic machinery of the chaperone [31]. The EM structure of Hsp90, in complex with Cdc37 and CDK4 at ~19 Å, was interpreted as showing the MD of Cdc37 sitting between the NTDs of Hsp90 and delivering CDK4, which engages with the NTD and MDs of Hsp90 [31] (Figure 2B). This structure may represent an early loading stage of the kinase complex. However, a higher resolution structure at 3.9 Å was presented using cryo-EM, which clearly represents a mature Hsp90–kinase complex (PDB 5FWM) [13]. In this structure, Hsp90 is in a closed conformation in which its NTDs are clearly dimerised (Figure 2C). Taking this higher resolution structure into account it is possible to interpret the lower resolution structure of this complex as being the same in terms of location of the individual domains. Nonetheless, it is clear that Cdc37 does engage with the lid-segment of the NTD of Hsp90 and that Cdc37 is able to inhibit Hsp90 ATPase activity. Thus, the spatial relationship of the proteins in an early-state complex and how the complex remodels to a later mature conformation remains to be confirmed.

In the higher-resolution structure or mature state, Cdc37 makes a number of interactions, including a very important interaction between a loop consisting of ^20^HPNI^23^ in Cdc37 and the C-terminal lobe of CDK4 (Figure 2D). Although the resolution of the cryo-EM map of this complex is variable, from 3.5 to 6 Å, some potential interactions can be inferred, but nonetheless caution should be exercised in these predictions. The extreme NTD of Cdc37 (residues 1 to 18) sits between the interface of the NTD and MD of one of the Hsp90 protomers, and probably helps to stabilise the n/m-domain association of Hsp90 (Figure 2E). Potentially the side-chain hydroxyl group of Tyr 4 from Cdc37 may be able to form a hydrogen bond with the side-chain amine group of Gln 128 of the N-terminal domain of the adjacent Hsp90 protomer. The side-chain carboxyl group of Gln 128 also appears close enough to form a hydrogen bond with one of the amine side-chain groups of Arg 392 (Arg 380 in yeast Hsp90) from the catalytic loop, where the same amine group is also engaged with the γ-phosphate of the bound ATP. Additional contacts between the catalytic loop of Hsp90 and Cdc37 residues Tyr 4, Trp 7 and Asp 8 may help stabilise the catalytic loop in its active conformation. The side-chains of Asp 8 and His 9 may also interact with those of Lys 36 and Asp 188 of the NTD of Hsp90. The next segment of the NTD of Cdc37, residues 10 to 18, interacts with the long helix of the MD of Hsp90 (residues Gln 385 to Ala 408) and potentially a series of salt bridges involving side-chains occur from the following interactions: Asp 14 and Glu 16 of Cdc37 with Arg 405 of Hsp90, Asp 17 of Cdc37 and Lys 402 of Hsp90 and Glu 18 of Cdc37 with Lys 399 of Hsp90 (Figure 2E). As seen with Aha1, the interaction of Cdc37 with the MD long helix of Hsp90 likely aids the conformational change of the catalytic loop to its open active state [5,32]. In addition, the side-chain of Glu 11 of Cdc37 may form a salt bridge with the side-chain of Arg 39 and phospho-Ser 13 (pSer 13) may form a bridging interaction by forming salt bridges with both His 33 and Arg 36 side-chains from the N-terminal end of the coiled-coil region of Cdc37 and with Lys 406 of the MD long helix of Hsp90. These interactions are critical for helping to position the conserved HPNI motif of Cdc37 for interaction with the C-lobe of Cdk4 (Figure 2D,E). The HPNI interaction mimics the interaction of the αC–β4 loop from the N-terminal lobe of CDK4 (HPNV in Cdk4 or HVNI in BRAF) (Figure 2D) and complex formation with Cdc37 is likely aided by the propensity of kinases to unfold [33]. Exiting from the coiled-coil segment of Cdc37, a beta strand then interacts with the 1AC β-sheet of Hsp90 (2CG9 nomenclature) and places the MD and CTD of Cdc37 on the opposite side of the Hsp90 dimer (see PDB 5FWL), which may be centred above Trp 312 (yeast Hsp90, Trp 300 and Hsc90, Trp 296) and Phe 341 (yeast Hsp90, Phe 329 and Yeast Hsc90, Phe 325). Interestingly, Trp 300 was previously identified as a client binding site [32] (Figure 2C).

The cryo-EM structure also reveals that the β4 and β5 strands of the n-terminal lobe of CDK4 are pulled apart and the CDK4 polypeptide threads through the centre of the Hsp90 dimer and positions the remaining N-terminal lobe on the adjacent Hsp90 protomer (Figure 2C). Interestingly, the structure from PDB 5FWL appears to show that the position of the remainder of the CDK4 n-terminal lobe is close to the long helix of the other MD of Hsp90. If this is the case, it appears that the NTD of Cdc37 may be able to influence the activation of one protomer of the Hsp90 dimer, while the client kinase may be aiding the other protomer and thus setting up hydrolysis of ATP by both Hsp90 protomers. Hydrolysis of ATP and dephosphorylation of pSer 13 by PP5 [34] may then destabilise the complex and release the kinase.

## 4. The Hsp90–Aha1 Complex

Aha1 remains the only potent activator of Hsp90 ATPase activity documented to date and is thought to be able to reduce the kinetic barrier presented by the rate-limiting conformational changes that Hsp90 has to undergo to produce a catalytically active unit [5,35]. Aha1 consists of an NTD separated from the CTD by a flexible ~60 amino acid linker. The structure of the NTD of Aha1 in complex with the MD of Hsp90 showed that Aha1 could modulate the catalytic loop of the MD of Hsp90 from an inactive to an active conformation, where Arg 380 (yeast Hsp90) could interact with the γ-phosphate of bound ATP [5]. The human NTD and the full-length structure of Aha1 were previously determined by NMR (PDB 7DMD and 7DME, respectively) [36].

Recent cryo-EM structures show that the activation of Hsp90 by Aha1 involves a multistep mechanism [8]. Binding of Aha1 to apo-Hsp90 leads to a partially closed Hsp90 dimer (EMD (Electron Microscopy Data Bank)-22238 and PDB 6XLB; 3.8 Å) (Figure 3A) that is bound by two molecules of Aha1. In this complex the NTD of Aha1 is engaged with a MD of Hsp90, as seen in the fragment crystal structure [35]. In contrast, each CTD of Aha1 is bound to the MDs of both Hsp90 protomers, and each contacts an amphipathic helix (residues Pro 324 to Asn 340) that was previously identified as a client–protein binding site [37]. The semi-closed conformation therefore represents a state in which the MDs are now closely associated and approach, but do not exactly match, the conformation seen in the yeast closed Hsp90–Sba1 structure [11]. To achieve this, a further 5° rotation of the Hsp90 MDs would be required. The binding of the CTDs of Aha1 appear to be incompatible with Hsp90 N/M-domain association, and the NTDs in the structure remained undefined. Binding of Aha1 to apo-Hsp90 therefore leads to the dissociation of the resting state for the N/M associated domains of Hsp90.

To achieve a fully closed state requires binding of nucleotide to Hsp90. With increasing concentrations of Aha1, in the presence of the non-hydrolysable nucleotide AMPPNP, a variety of Hsp90–Aha1 complexes were seen, with the equilibrium shifting from Hsp90 bound with one Aha1 molecule (with only the CTD being visible; HAc; EMD 22240 and PDB 6XLD; 3.66 Å) through to a Hsp90 complex with two Aha1 molecules bound, but with only the CTD visible (HAcc; EMD-22241 and PDB 6XLE; 2.74 Å) and finally to an Hsp90 complex in which two Aha1 molecules are bound, with two CTD and one NTD visible (HAncc; EMD-22242 and PDB 6XLF; 3.15 Å). In the HAncc structure, Hsp90 is now N-terminally dimerised and Aha1 binding has been restructured and forms a tighter bound complex. The NTD of Aha1 is found to be tilted by 30° relative to the state with apo-Hsp90, and consequently, the original interface with the MD of Hsp90, as seen in the fragment structure, is broken (Figure 3B) and new interactions are formed. Specifically, Aha1 residues 1 to 10 containing the conserved motif NxNNWHW are found bound across the dimer interface and appear to stabilise n-terminal dimerisation (Figure 3C). The two conserved Trp residues of the NxNNWHW motif also form stabilising interactions, where Trp 9 is engaged with the lid segment of one of the Hsp90 protomers and Trp 11 is engaged with the MD of the same Hsp90 protomer. Deletion of the n-terminal segment (residues 1 to 11) of Aha1 has previously been shown to reduce its ability to activate Hsp90 [38]. The interaction of the n-terminal segment of Aha1 is further supported by a small helix coil that binds over the n-terminal segment of Aha1 but also forms stabilising interactions with the lid segment of Hsp90, and Tyr 165 appears to be important in this interaction (Figure 3C).

In addition to the changes that occur between the NTD of Aha1 and Hsp90, the interaction of the CTD of Aha1 also alters its interaction with Hsp90. The Aha1 residues Phe 264, Asn 267, Asn 268, Leu 287, Arg 327, Asn 331 and Tyr 335 no longer interact with the amphipathic loop (residues Pro 324 to Asn 340) but shift their interaction to a loop formed by residues Ser 297 to Leu 304 (Figure 3D). The shift in conformation from the apo-Hsp90 to the AMPPNP bound structure involves a 15° counter clockwise rotation as well as a deep pocket within the Aha CTD opening up to accommodate Trp 296 of Hsp90 (yeast Hsp90 Trp 300 and Human Hsp90b Trp 312), having switched from interacting with Phe 328 (yeast Hsp90 Phe 332 and human Hsp90β Phe 344). Trp 296 (yeast Hsp90 Trp300) is the same residue identified as a client–protein binding site [32], and also involved in interactions with CDK4 in the Hsp90–Cdc37–Cdk4 cryo-EM structure, discussed above. It would therefore appear that there is a series of aromatic amino acid residues (Yeast Hsp90 Trp 300, Phe 329 and 332; Yeast Hsc90 Trp 296, Phe 325 and 328 and human Hsp90β Trp 312, Phe 341 and 344;) that are involved in both client protein and co-chaperone binding (Figure 3A). Mutating yeast Trp 296 to either Ala or Gly has been shown to significantly reduce the ATPase activity of human Hsp90β, but the equivalent mutation in yeast Hsp90 (Trp 300A) did not affect its ATPase activity [8,32]. It therefore appears that interaction with these aromatic residues by co-chaperones and clients may act as a signal to regulate Hsp90 ATP hydrolysis by communicating the co-chaperone and client protein–bound state in the complex.

Another cryo-EM structure of Hsp90–Aha1 in complex with the slowly hydrolysable ATPγS (EMD-22243 and PDB 6XLG) contains a dimer of Hsp90 and two Aha1 molecules, where one NTD and two CTD are visible (HAnccg). This is equivalent to the AMPPNP structure discussed above, except that the Hsp90 protomer whose MD is bound by the NTD of Aha1 has hydrolysed the ATPγS to ADP and the other protomer still retains intact ATPγS. Collectively, these structures allowed the authors to propose a conformational cycle involving four steps towards hydrolysis of ATP by Hsp90 [8] (Figure 3E). Initially, Aha1 can be recruited to apo-Hsp90 by binding of its NTD to the MDs of Hsp90. This would produce a complex equivalent to the fragment structure determined by crystallography [35]. Next, the semi-closed Hsp90 state is formed by the binding of the CTDs of Aha1 to the MDs of Hsp90, which also displace the NTDs of Hsp90. However, the NTDs of Hsp90 are now primed for ATP binding and dimerisation as well as for Hsp90 N/M-domain association. ATP then binds and this signals a restructuring of the Hsp90–Aha1 complex to allow Hsp90 NTD dimerisation. Finally, Aha1 reorganises and helps stabilise the N-terminally dimerised state of Hsp90 and this in turn signals for ATP hydrolysis, which in the presence of a single molecule of Aha1 appears to occur sequentially for each protomer of Hsp90. The multistep activation of Hsp90 by Aha1 is consistent with the findings from an NMR study that found a two-step binding mechanism for Aha1 and that structural changes were induced near the ATP binding site of Hsp90, which conspire to activate Hsp90 [8]. Although, another model suggests that a single Aha1 molecule bound to Hsp90 can cause asymmetric activation of Hsp90 and thus fully activate it [21]. However, the Hsp90 ‘sequential ATP hydrolysis’ model does not take into account the presence of client protein. Thus, the co-ordinated interaction of a client on one protomer of the Hsp90 dimer and a co-chaperone on the other might allow the simultaneous hydrolysis of ATP by both protomers of Hsp90. Clearly, further work to determine the exact timing of ATP hydrolysis by each protomer of an Hsp90—co-chaperone—client protein complex is therefore required.

## 5. The Hsp90–p23–FKBP51 and Hsp90–Sba1 Complex

Crystal structures of human FKBP51 (PDB 5OMP) and yeast Sba1 (PDB CG9) (p23 in humans) have been reported [11,39,40]. FKBP51 is an immunophilin and possesses peptidyl propyl isomerase (PPIase) activity, which catalyses the cis-trans isomerisation of proline and includes members such as FKBP52 and the cyclophilin Cyp40 [41]. FKBP 51 contains three distinct domains, a catalytically active N-terminal PPIase domain (FK1), an inactive PPIase MD (FK2) and a C-terminal tetratricopeptide (TPR) domain. FKBP51, FKBP52 and p23 have been reported as members of steroid hormone complexes [42,43], where they facilitate client protein activation and localisation [44,45,46]. FKBP52 was shown to potentiate the GR receptor activation when hormone levels where limiting, whereas FKBP51 appears to block potentiation [45]. p23 is a CS domain–containing co-chaperone that appears to enter the Hsp90 activation cycle in its late stage and, as with FKBP51, favours the closed-nucleotide bound conformation of Hsp90 [11,39]. Sba1 in yeast appears to down regulate the ATPase activity of Hsp90 and stabilises the closed Hsp90 complex [11].

The role of p23 in Hsp90 complexes was previously determined in the context of steroid hormone activation, which has been elegantly reviewed previously [43,47,48]. The work described by these authors is relevant not only to this section but also to the subsequent sections that look at the loading and maturation complex for the glucocorticoid receptor (GR). Much of the work detailed by Pratt, WB, Toft, D, O and their co-authors showed that a Hsp90/Hsp70-based chaperone complex was responsible for regulating the steroid binding, trafficking and turnover of GR. An ATP-dependant activation cycle involving HOP, p23, Hsp40, FKBP51/52 and the Hsp90/Hsp70 complex is able to assemble ligand-binding domain (LBD) of the GR with Hsp90, in which the hydrophobic ligand-binding cleft is opened to allow access for steroid hormone binding. Much of this work has now been confirmed by the advances discussed below.

The crystal structure of Sba1 was determined in complex with Hsp90 (PDB CG9) [11] and is essentially similar to the cryo-EM structure of human p23 in complex with Hsp90 (PDB 7KRJ) [39,40]. Essentially, Sba1 is bound between the NTDs of N-terminally dimerised Hsp90 (Figure 4A) and contacts the closed lid-segment, the NTDs of both protomers and the MD of one Hsp90 protomer. Some critical interactions can be seen, which include a hydrogen bond between the side-chain of Asn 97 of p23 and the carbonyl main-chain of Leu 122 in the lid-segment of Hsp90, a salt bridge from the side-chains of Lys 95 of p23 and Glu 336 in the MD of Hsp90 and a hydrogen bond between the side-chains of Asn 104 of p23 and Asn 35 of Hsp90. Further interactions occur between the side-chain of Arg 71 of p23 and those from Ser 31 and Glu 22 of Hsp90 (Figure 4A). Collectively, the lid-segment, N-terminal dimerisation and n/m-domain association of Hsp90 are all stabilised. This is not too dissimilar to the action of Aha1, although for Sba1/p23 the ATPase activity is downregulated [35,49]. However, what is clear from these structures is that the binding of Aha1 and p23 to the same side of Hsp90 is incompatible (Figure 4A). A further set of interactions that is worthy of a mention is the interaction between the conserved yeast Sba1 residues Phe 121 (human Phe 103) and Trp 124 (human Trp 106), which sit in a hydrophobic pocket formed by Leu 315, Asp 373, Leu 376, Gln 385, Lys 387 and Val 391 of yeast Hsp90. These interactions may help stabilise the open conformation of the catalytic loop so that Arg 380 (yeast Hsp90) is able to interact with the γ-phosphate of bound ATP (Figure 4B). In fact, some analogies can be drawn between the Sba1 and Aha1 interactions with the MD long helix (residues Lys 399 to Ala 420) of yeast Hsp90 (Figure 4C). When comparing the interactions of the conserved Phe 121 (human Phe 103) and Trp 124 (human Trp 106) of Sba1 to those of Aha1, an analogous set of interactions occurs, where Aha1 residues bind into the same hydrophobic pocket(s) as those of Sba1. The interaction with Aha1 involves Ile 64, Leu 66 and Trp 11 (from the conserved NxNNWHW motif). Comparing the various structures of the human and yeast Hsp90–Aha1 complex, there is a gradual engagement of these residues as the closed conformation of Hsp90 is formed (Figure 4C). The least engaged situation is with the apo-Hsp90, in which two Aha1 molecules are bound (EMD-22238 and PDB 6XLB) and then closer engagement is seen in the yeast Aha1 fragment structure (PDB 1USU) and finally in the human AMPPNP bound structure (EMD-22242 and PDB 6XLF), where Aha1 has been tilted to fully engage with the NTD of Hsp90 (Figure 3B,C and Figure 4C). The engagement of Sba1 and Aha1 with the MD long helix likely helps to stabilise the catalytic loop of the MD and N/M-domain association.

There are many structures reporting the interaction of the conserved MEEVD motif of Hsp90 with TPR domain–containing proteins [50,51,52,53]. Recently, the cryo-EM structure of intact FKBP51 in complex with Hsp90 and p23 was reported (PDB 7L7I) [39]. The structure shows the expected TPR domain interaction with the MEEVD motif of Hsp90 but also an unexpected interaction with the C-terminal helix 7 extension of the TPR domain. The helix is kinked compared to the crystal structure of FKBP51 [54] and docks in a small hydrophobic cleft at the dimer interface at the C-terminal end of the CTD of Hsp90 (Figure 4D). Consequently, the stoichiometric binding of FKBP51, where helix 7 is also bound to dimeric Hsp90 is 1:1 [FKBP51: Hsp90 dimer]. FKBP51 helix residues Ile 408, Tyr 409, Met 412, Phe 413, Phe 416 and Ala 417 become unfolded and interact with Hsp90 residues that line the hydrophobic cleft (Leu 694, Lys 657, Ser 658, Asp 661, Arg 690, Met 691, Ile 692 and G695). Interestingly, Tyr 409, Met 412 and Phe 413 appear to be conserved among other immunophilin proteins. Additionally, the side-chain of Arg 690 from each Hsp90 protomer forms a salt bridge to the side-chains of Asp 405 and 420. Interactions are also seen between Hsp90 and residues on the H5–H6 loop of FKBP51, including the carboxyl main-chain of Asn 365 of FKBP51, which makes contact with the side-chain of Asn 655 in Hsp90 and possibly the side-chain of Asp 366 of FKBP51 with that of Lys 657 of Hsp90. Finally, some minor contacts are visible between the FK1 domain of FKBP51 and Hsp90 along the MD and adjacent to the substrate-binding loops. The affinity for the binding of FKBP51 was shown to be higher for the closed n-terminally dimerised state of Hsp90, and docking of the helix 7 extension with Hsp90 appears to be specific for the closed conformation of Hsp90 [39]. The catalytically active FK1 domain appears to be placed with its active site cleft facing the MD client–binding residues and modelling studies suggest that Pro 173 from CDK4 may be accessible to the active site of the FK1 domain [55]. Pro 173 is part of the highly conserved APE motif found at the base of the activation loop of kinases. Non-canonical APE motifs, such as AAE in ARAF, lead to a lower allosteric and catalytic activity as a result of a lower propensity to undergo homodimerisation, showing the importance of this motif in attaining an active state conformation [56]. The activation loop, including the APE motif, is very flexible [56,57] and the action of PPIase activity by immunophilins such as FKBP51 on the conserved proline residue may promote assembly of the activation segment into an active state. However, the mechanistic details that may link this to kinase dimerisation and activation, whether through allosteric activation and cis-autophosphorylation or by trans-phosphorylation [57,58,59,60,61], are yet to be established. However, it was also suggested that with particular clients, such as the glucocorticoid receptor (GR), that the FK1 domain may provide a scaffolding function and thus activation of GR is independent of the PPIase activity of the FK1 domain [39].

## 6. Hsp90–Hop–Hsp70–GR Loading Complex

Another client chaperone complex that was determined by cryo-EM is the loading complex for the glucocorticoid (GR) client with Hsp90, Hop and Hsp70 at 3.6 Å (PDB 7KW7) [62], which is thought to represent an early loading complex. The structure revealed some unexpected findings. Namely, two Hsp70 molecules were found in the complex, the first chaperoning the GR client and the second interacting with Hop, which itself was found to interact with all components within the complex. Within this structure, Hsp90 appears to adopt a semi-closed conformation, where the NTDs are oriented correctly but remain posed for dimerisation. The lid-segments of the NTDs remain open and are devoid of nucleotide, and the n-terminal β-strand, which can undergo strand exchange with the adjacent protomer, remains in place on its own NTD. The two Hsp70 molecules are bound by ADP and adopt an ADP-like conformation that is similar to that of the co-crystal structure (PDB 3AY9) [63]. Both Hsp70 molecules interact with Hsp90 in almost identical ways and there are two major interfaces between each Hsp70 molecule and its associated Hsp90 protomer. In the first heterodimeric interface a β-strand from the outer face of the MD β-sheet of Hsp90 inserts itself into the cleft between the subdomains IA and IIA of Hsp70 (Figure 5A). This cleft is only available in the ADP state of Hsp70. Side-chain interactions, within the first interface are seen between Lys 414, 418 and 419 of Hsp90 with the side-chains of Asp 213, 214 and 218 of Hsp70, respectively. In addition, the main-chain carbonyl of Asp 214 of Hsp70 makes a hydrogen bond with the side-chain Gln 334 of Hsp90, the main-chain amide of Gln 334 is hydrogen bonded to the carboxyl main-chain of Gly 215 of Hsp70 and the side-chain of Glu 332 of Hsp90 makes a set of interactions with the side-chains of Asn 174 and Thr 177 from Hsp70. Similarly, Arg 171 of Hsp70 makes side-chain interactions with Glu 336 and Asp 393 of Hsp90. In addition to these polar interactions there are several hydrophobic residues, Leu 334 and Val 411 from Hsp90 and Ile 216 and Phe 217 from Hsp70, which help to stabilise the interface. In the second interface between Hsp70 and Hsp90, Arg 60 and Tyr 61 of Hsp90 make side-chain interactions with Asp 160 from Hsp70 (Figure 5A). However, the presence of Hsp70 is incompatible with the closed N-terminally dimerised state of Hsp90, and transition to the closed state likely requires nucleotide exchange by an Hsp70 nucleotide exchange factor such as Bag1 in order to advance the complex.

Within the Hsp90–Hsp70–Hop–GR complex, only three domains of Hop were visible, TPR2A, TPR2B and the DP2 domain. These domains appear to be fully sufficient for full GR activation [64,65]. The TPR2A and TPR2B domains are bound by the conserved C-terminal -EEVD extensions of Hsp90 and Hsp70, respectively, and subdomain IIA of Hsp70 is found to be critical in positioning the relative positions of Hsp90 and HOP (Figure 5B). All three visible domains of HOP make contact with Hsp90 such that the Hsp90 protomers are fixed in their semi-closed conformation. Perhaps the most interesting interactions between HOP and Hsp90 are the interactions of the DP2 domain with two residues (Trp 606 and Met 614 in Hsp90a or Met 550 and Phe 554 in HTPG) previously identified in a set of conserved client-protein binding-site residues in HTPG (Glu 466, Trp 467, Asn 470, Met 546, Met 550, Leu 553 and Phe 554 and in human Hsp90α these are Glu 527, Tyr 528, Gln 531, Trp 606, Met 610, Ile 613 and Met 614, respectively) (Figure 5C) [66]. Collectively, a hydrophobic set of interactions involves Met 610, 614, 625 and 628, Ala 618 and Trp 606 from Hsp90 and Pro 502, Ala 503, Leu 506 and Ile 507 from DP2. In addition, a hydrogen bond is formed between the side-chains of Thr 624 of Hsp90 and Asp 501 from DP2 (Figure 5C).

Reminiscent in the way CDK4 is unfolded and passes through Hsp90 in the Hsp90–Cdc37–Cdk4 complex, the n-terminal segment of the GR domain threads through the lumen of Hsp90 (Figure 5D). While GR is poorly structured, some important interactions with Hsp90 can be inferred and include Trp 320 and Phe 349. In particular, Trp 320 (yeast Hsp82 Trp 300) was described as not only interacting with the GR (Figure 5D) but also with the DP2 domain of HOP and Aha1 as well as with the kinase client CDK4. Other potential interactions include the side-chain of Asp 626 of the GR with that of Thr 603 of Hsp90, the main-chain carbonyl of Lys 703 of the GR and the side-chain nitrogen of Lys 410 of Hsp90 and main-chain nitrogen of Lys 410 of the GR with the side-chain oxygen of Gln 405 of Hsp90 (Figure 5D). Residues from Ile 539 to Ser 550 of the GR pass through Hsp90 and connect to the GR ligand-binding domain (LBD) helix 1 that is cradled by a hydrophobic cleft in the DP2 domain of HOP (Figure 5E). The hydrophobic residues from the GR helix 1 include Leu 532, 533, 535 and 536, Val 538 and Ile 539 (conserved motif L^532^XXLL^536^) and residues forming the hydrophobic pocket of the DP2 domain include Met 499, Arg 505, Leu 508, Gln 512 and Leu 534 (Figure 5E). Finally, the upstream amino acid residues representing pre-helix-1 (residues Ala 523 to Thr 531) lead to Ser 519 through to Leu 525, which are bound within the substrate binding site of Hsp70 (Figure 5E). Overall, this complex holds the GR in an inactivated state, as is consistent with observations suggesting that Hsp70 inactivates hormone binding by the GR and that Hsp90 eventually restores its activity [67]. The loss of specific co-chaperones, such as Hsp70 and HOP, from the complex is therefore required for the maturation of the GR to the active state.

## 7. Hsp90–p23–GR Maturation Complex

The cryo-EM structure of the Hsp90–p23–GR maturation complex was recently described and was determined at 2.56 Å (PDB 7KRJ) [40]. The Hsp90–p23 components are essentially similar in structure to that previously described for the nucleotide-bound yeast Hsp90–Sba1 structure, except that a single p23 molecule is bound on the same side as the bound GR. the GR is now remodelled from its loading complex and held in an active state conformation. With the loss of Hsp70 and HOP, helix 1 (Leu 532 to Ile 539) and the pre-helix-1 residues (Ala 523 to Thr 531) have retracted back towards the GR LBD (ligan binding domain). The pre-helix-1 is now held within the hydrophobic lumen of Hsp90 (Figure 6A). Leu 525 of the GR faces a hydrophobic pocket lined by Ile 525, Tyr 528, Leu 447 and His 450 and by Leu 619 from the other Hsp90 protomer. Leu 528 of the GR makes a similar set of interactions as Leu 525, but with the symmetry related residues (Ile 525, Tyr 528, Leu 447, His 450 and Leu 619) of the Hsp90 dimer. Some additional polar interactions occur between the carbonyl main-chain of Ala 523 of the GR and the side-chain amide group of His 450 of Hsp90, while the carbonyl main-chain of Thr 529 of the GR makes the symmetry related contact with the side-chain of His 450 in the other Hsp90 protomer (Figure 6A). The residues involved in the pre-helix interactions with Hsp90 were previously identified (HTPG, Glu 466, Trp 467 and Asn 470; Hsp90α, Glu 527, Tyr 528 and Gln 531 and Yeast Hsp90, Glu 507, Tyr 508 and Thr 511) or are close to amino acid residues forming a client-protein binding site, together with residues from an amphipathic helix (HTPG, residues Met 546 to Ala 555; Hsp90α, residues Trp 606 to Lys 615 and Yeast Hsp90, Trp 585 to Lys 594) [37]. The GR LBD also interacts with residues Trp 320 and Phe 349 from the adjacent Hsp90 protomer (Figure 6B). The side-chain of Trp 320 potentially forms a polar interaction with the main-chain carbonyl of Asn 586 and is also shielded by His 588 of the GR. In contrast, Phe 349 points towards a hydrophobic pocket lined by Gly 583, Asn 586, Leu 685, Ile, 689 and Thr 692. Finally, a potential polar interaction between the side-chain oxygen of Asn 586 of the GR and the side-chain of Arg 346 is seen (Figure 6B).

The remodelling of the GR in the Hsp90–GR–p23 complex sees helix 1 packing against the amphipathic helix of Hsp90 (Hsp90α, residues Trp 606 to Lys 615) and helixes 8 and 9 of the GR LBD (Figure 6C). Trp 606 of Hsp90 forms the hub of the hydrophobic interaction with the GR, packing up against Val 538 of helix 1 of the GR and the amphipathic helix residues of Hsp90, Met 610 and 614 (Figure 6C). Met 628 of Hsp90 is also packed between Leu 535 and Val 538 of helix 1 of the GR LBD. Finally, the side-chain carboxyl group Glu 537 from helix 1 forms a bipartite polar interaction with the side-chains of Gln 531 and Lys 534. Collectively, these interactions allow the GR LBD to adopt an active conformation, where helix 12 is in the agonist-bound state [68], and density is visible for an agonist, presumed to be dexamethasone present from the purification of the GR.

As seen with the yeast Hsp90–Sba1 structure [11], the C-terminal tail of p23 interacts with the long helix of the MD of Hsp90 and the catalytic-loop Arg 400 (yeast Hsp90 Arg 380) is in contact with the bound nucleotide (Figure 6D). Specifically, Phe 103 of p23 points towards a hydrophobic pocket lined by Leu 335, Asp 393, Pro 395 and Ile 408 of Hsp90. In contrast, Trp 106 of p23 sits between Leu 335, Lys 407, Ile 408 and Val 411 of Hsp90. In addition, the side-chain of Trp 106 of p23 forms a polar interaction with the side-chain of Gln 334 of Hsp90 (Figure 6D). Finally, the side-chains of Asp 108 and Asp 111 of p23 form salt bridges to the side-chain of Lys 414 of Hsp90. A relay of charged or polar interactions continues between the side-chains of Asp 112, 114 and 116 of p23 with Ser 708, Asn 171 and Lys 695 of the GR, respectively (Figure 6E). The extreme C-terminus of p23 then ends with a helix that interacts with the GR (Figure 6F). The side-chain of Asp 133 of p23 forms a polar interaction with the side-chain of Gln 713 of the GR. A series of hydrophobic residues from p23 (Met 117, Phe 123, Met 126, 127 and 130) are in hydrophobic interaction with Ser 709, Trp 712, Gln 713, Phe 715 and Tyr 716 of the GR (Figure 6F). Phe 123 and Met 127 of p23 also appear to stabilise the C-terminal tail of the GR by interacting with Phe 774 and His 775 (Figure 6G) and potentially helping to stabilise helix 12 of the GR in its agonist binding state. This study ultimately identified a conserved motif in the C-terminal tail of p23, FxxMMNxM, which was also identified in the coactivator 3 (NCoA3), which functions as a co-activator of steroid hormone receptors. In vitro ligand-binding studies showed that deletion of p23 residues beyond Asp 133 did not affect chaperone-mediated ligand binding. However, deletion of residues Ser 113 and beyond resulted in a reduction of such activity [40]. This suggested that other core components of the p23 interaction with Hsp90 are also important for GR activation. In contrast, in vivo studies showed that both the mutants reduced levels of activated GR, which may suggest that p23 has a more dominant downstream function after ligand binding.

## 8. The GR Activation Cycle

Collectively, the structures of the loading and maturation complex of the GR, as well as numerous biochemical studies, suggest a series of steps that lead to the activation of the GR [40,62,67,69]. The first step in the activation cycle of the GR is the stabilisation and inactivation of the GR by Hsp70 and capture of the GR is dependent on Hsp40 and ATP hydrolysis by Hsp70 [70,71,72] (Figure 7). In this state, the ligand bound to the GR would be released. Meanwhile, the binding of HOP to Hsp90 preassembles Hsp90 in order to receive the Hsp70–GR client complex [62,69,73]. The main interaction of HOP is with the MD–CTD junction of Hsp90 and prevents its rotation to a conformation that would allow Hsp90 N-terminal dimerisation [69]. Specifically, the TPR1 domain of HOP sterically blocks the N-terminal dimerisation of Hsp90 by binding between the Hsp90 monomers, while simultaneously interacting with the adjacent MD and CTD of Hsp90. Step 2 of the activation cycle was captured by the cryo-EM structure of the Hsp90–Hsp70–HOP–GR complex [62]. The structure reveals how the Hsp70–GR client complex initially associates with the preassembled Hsp90–HOP complex. In this complex the pre-helix-1 segment of the GR is captured by Hsp70 and helix 1 stabilised by the DP2 domain of HOP (Figure 5E). Interaction with Hsp90 allows the GR post-helix-1 segment to thread through a semi-closed Hsp90. The NTDs of two Hsp70 molecules that were found bound to Hsp90 interact with the NTD of Hsp90, thus providing a link between the ATPase activity of Hsp70 and Hsp90, where the Hsp90 lid-segments are close to the Hsp70 interaction interface with Hsp90. In this state, the GR is held in an inactive conformation (Figure 7). In step 3, ATP binding to Hsp90 and its hydrolysis leads to the release of Hsp70 and HOP from the complex [67]. Evidence suggests that this is a direct result of the hydrolysis of ATP by Hsp90, rather than direct action by a nucleotide exchange factor such as Bag-1 acting on Hsp70. Instead, Bag-1 may play a role in stalled Hsp90–client complexes [67]. It was suggested that during the transition from the Hsp70-present to the Hsp70-absent complex with Hsp90, hormone could bind to the GR LBD. The product of step 3 is captured by the Hsp90–p23–GR cryo-EM structure [40], which is able to bind cortisol. This structure shows how remodelling of the GR allows pre-helix-1, previously held by Hsp70, to move into the lumen of the fully closed Hsp90 molecule and simultaneously may allow helix 1 to associate with the GR LBD and seal the hormone binding pocket. This suggests that the hormone needs to have already bound to the GR. Within this complex, p23, the binding of which is favoured by the fully closed conformation of Hsp90, appears to facilitate the activation of the GR. Firstly, it stabilises the closed conformation of Hsp90, and secondly, it plays a role in stabilising the dynamic helix 12 of the GR in its agonist-binding conformation [40]. In step 4, activated GR must be released from the complex, but it remains unclear what the trigger for this is. However, results show that the activation of the GR is enhanced by the presence of p23 in a fully reconstituted chaperone system that contains Hsp90, Hsp70, Hsp40, HOP and the GR LBD. This suggests that the action of p23 appears to occur prior to the GR–helix 1 capping of hormone access to the LBD of the GR. This may indicate that release of the GR from the Hsp90 complex occurs as a direct result of hormone binding [67]. Clearly, some finer details of the cycle need to be established.

## 9. Hsp90–Tau Complex and Hsp90–FKBP51–Tau Complex

Tau stabilises microtubules that serve as tracks during axonal transport within neurons [74]. Hsp90 can help stabilise and bring about proteasomal degradation of Tau [75]. Unlike CDK4 and the GR LBD, Tau is classified as a partially unfolded protein. Hsc70 can associate with Tau, specifically recognising two motifs (^275^VQIIN^279^ and ^306^VQIVY^310^), and bring about some core domain folding, but significant regions of Tau remain unfolded [76]. Hsp90 has been shown to recognise Tau directly, without co-chaperone delivery [77], which may reflect the fact that Tau is significantly unfolded. The *K*_D_ for Hsp90–Tau association was determined to be approximately 4.8 μM and ATPγS did not alter binding affinity [77].

Using an NMR approach, together with small angle scattering, a structural model was obtained for the interaction between Hsp90 and Tau [77]. Hsp90 was found to recognise a broad region of Tau, including its aggregation-prone repeats and the Hsc70 binding motifs. Specifically, regions ^226^VAVVRT^231^, ^244^QTAPV^248^, ^275^VQIINK^280^, ^306^VQVYK^311^, ^340^KSEKL^344^ and ^377^TRFEN^381^ are bound and, interestingly, these motifs have been implicated in Alzheimer’s disease [78]. Collectively, these regions represent hydrophobic centres with a propensity for a positive net charge, although it was also noted that negative charges were not wholly excluded. It was concluded that it is these specific properties of Tau, rather than its unfolded nature, that are the contributing factor for Hsp90 binding [77], although access to such motifs is obviously paramount. The distribution of the hydrophobic Tau residues appears to resemble those of the exposed residues in the intermediates of folding Tau.

By employing specific isotope labelling of isoleucine methyl side-chains and methyl transverse relaxation–optimised spectroscopy (methyl-TROSY) a subset of Hsp90 residues (Ile, 20, 74, 90, 369 and 440) were implicated in the binding of Tau, indicating a broad Hsp90 binding surface that included both the NTD and MD of Hsp90. Together with Tau, ATPγS was seen to modulate the dynamics of Hsp90 and that the presence of Tau breaks Hsp90 symmetry, as seen with other client protein complexes (CDK4 and the GR). The most intense binding areas of Hsp90 included residues in both the NTDs and MDs of Hsp90, including Leu 24 and 27, Phe 32, Ile 105 and Ala 106 in the NTD and Leu 388, Ile 390, Phe 344, Lys 406 and 410 and Thr 446 in the MD of Hsp90 (Uniprot entry P08238 and numbering from Met 1). These hydrophobic residues compliment interactions between charged residues of Hsp90, including Glu 81 and 393 and Asp 367 and 518 and positive charged segments within Tau. The extensive binding region of Hsp90 appears to allow a high number of low affinity contacts with Tau.

Another methyl-TROSY study looked at the Hsp90–FKBP51–Tau complex [79]. Association of FKBP51 with the Hsp90–Tau complex can promote amorphous aggregation of Tau [80,81,82]. Hsp90 residues affected by FKBP51 titration appear to include the loop around the catalytic arginine in the MD of Hsp90 but also the residues that appear to line the internal surface of the Hsp90 dimer. This is in contrast to the FKBP51 interactions seen in the closed Hsp90–p23–FKBP51 complex [39]. All domains of FKBP51 were implicated in binding the open conformation of Hsp90. Essentially, Hsp90 and FKBP51 are arranged in a head-to-head topology, where the NTD, MD and CTD of Hsp90 interact with the FK1 domain, the FK1 and FK1-TPR domains and the FK2-TPR domains of FKBP51, respectively. Significantly, the NTD of Hsp90 is rotated away from the catalytic residue of the MD of Hsp90 and the catalytic PPIase pocket of the FK1 domain is solvent accessible. The study also showed that FKBP51 binding helps to stabilise the Hsp90-Tau complex and that Tau’s proline-rich region clusters close to the catalytically active FK1 PPIase domain. These proline-rich segments are also sites of Tau phosphorylation, and it was proposed that alteration of the normal proline isomerisation of Tau could lead to enhanced oligomerisation and an increased susceptibility to Alzheimer’s disease [80].

## 10. Concluding Remarks

X-ray crystallography and biochemical studies have identified an Hsp90 ATPase-driven catalytic cycle that is essential for the activation and maturation of client proteins. The role of a variety of co-chaperones was systematically determined and a variety of co-chaperone and client–protein binding sites on Hsp90 were identified, which advanced our understanding of the regulation of the cycle. However, the mechanistic details by which client proteins were activated remain enigmatic due to the structural complexity of Hsp90 client protein complexes. Thus, a unified mechanism of action was not easy to establish.

Recent advances in cryo-EM have now enabled us to understand how Hsp90 recognises client proteins and how it brings about their activation or maturation mechanistically. It appears that clients such as the GR and CDK4 are structurally dynamic and prone to aggregation. These unstable conformations can be captured by Hsp70 or co-chaperones such as Cdc37, stabilising their conformations and allowing their delivery to Hsp90. Other co-chaperones may preassemble Hsp90 so that it is competent for client protein binding and this appears to be a role that HOP plays in the activation of the GR. Other clients, such as Tau, that are inherently unfolded appear to bypass a co-chaperone loading stage, as their binding conformation is accessible to Hsp90. However, this does not necessarily exclude the possibility that Hsp90 might be preassembled for their interaction by a yet undefined co-chaperone. Ultimately, the Hsp90 cycle remodels the client protein towards a state that leads eventually to its activation, either by binding a small molecule hormone, as in the case of the GR, or by refolding, as in the case of kinases.

A series of aromatic residues on Hsp90 appear to play important roles in client protein and co-chaperone recognition that may also be important for communicating the bound state of the Hsp90 complex to features of Hsp90 that carry out ATP hydrolysis. Thus, these aromatic residues may link the presence of client or co-chaperones in the Hsp90 complex to the ATPase activity of Hsp90. Finally, the advances in describing the mechanistic details by which client proteins are activated by Hsp90 will increase our understanding of the underlying mechanisms of disease caused by the dysregulation of the Hsp90 client–protein system. It is clear that we are now entering into a new era for Hsp90 research that will see novel ways of targeting Hsp90-associated disease.

## Figures and Tables

**Figure 1 biomolecules-12-00600-f001:**
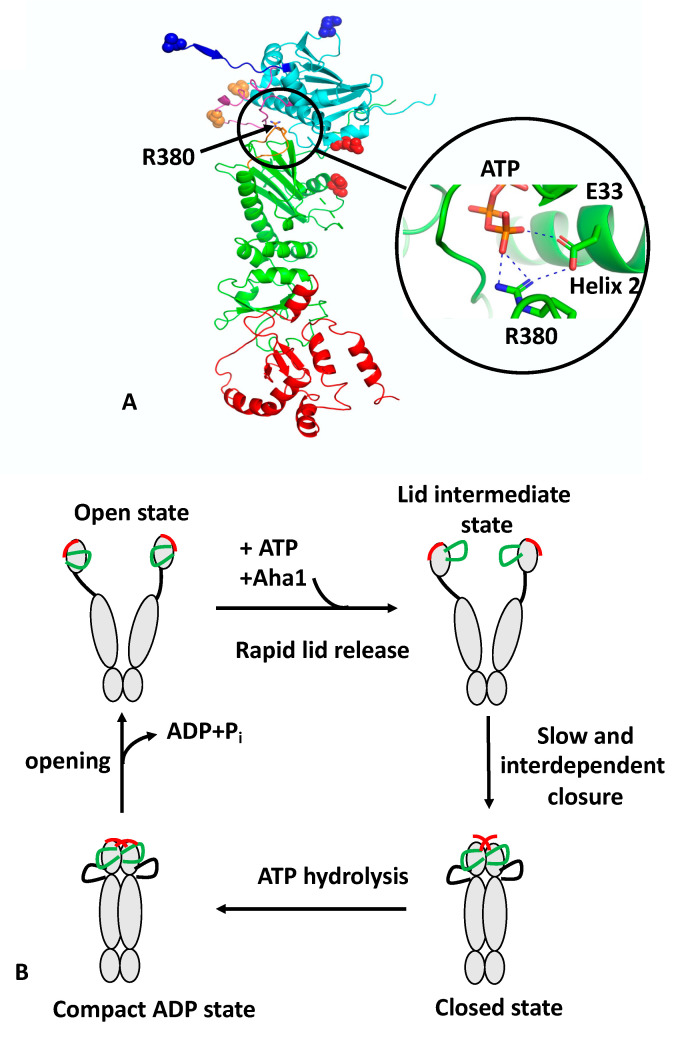
PyMol cartoon showing the structure and conformational switches of Hsp90 and its catalytic cycle. (**A**) Monomeric Hsp90 in the active, N-terminally dimerised closed conformation. Hsp90 consists of three domains: cyan, NTD; green, MD; and red, CTD. Four structural features are shown: blue, the structure involved in the N-terminal β-strand cross-subunit swapping; magenta, the lid-segment involved in lid closure; cyan and green, the NTD and MD domains involved in intra-subunit association; and magenta, the catalytic loop containing Arg 380 (shown as sticks). Attachment points for the fluorophore or Trp are shown as residues in sphere representation: blue, represents the n-terminal β-strand cross-subunit swapping switch pair (normally as a pair between both protomers); red, the NTD and MD intra-subunit association switch; and gold, the lid-closure switch. The close up shows the interaction of Arg 380 with the γ-phosphate of ATP and Glu 33. Dotted blue lines represent salt bridges. (**B**) The catalytic cycle of Hsp90. The apo state of Hsp90 occupies a heterologous ensemble of open-conformers. The lid (green) and n-terminal β-strand (red) are highly mobile structural elements showing a sub-millisecond reconfiguration time. The binding of ATP to the NTD of Hsp90 leads to a rapid release of the lid segment to an intermediate conformational state, between the so-called open- and closed-lid states. The association of Aha1 can remodel the lid but also preassociates the n- and m-domains for accelerated closure. Cooperative action of conformational switches leads to full closure of the molecular clamp. This involves closure of the lid over the ATP-binding pocket, a cross-subunit swap of the β-strands, and association of the n- and m-domains, which collectively are slow and interdependent. The β-strand swapping is weakly coupled with the other motions. The cycle completes following the hydrolysis of ATP, which leads to a compact, ADP-bound conformation. The hydrolysis of ATP appears to be non-cooperative between the two protomers of Hsp90. Following ATP hydrolysis, Hsp90 then relaxes to an open state, with concomitant release of ADP and inorganic phosphate.

**Figure 2 biomolecules-12-00600-f002:**
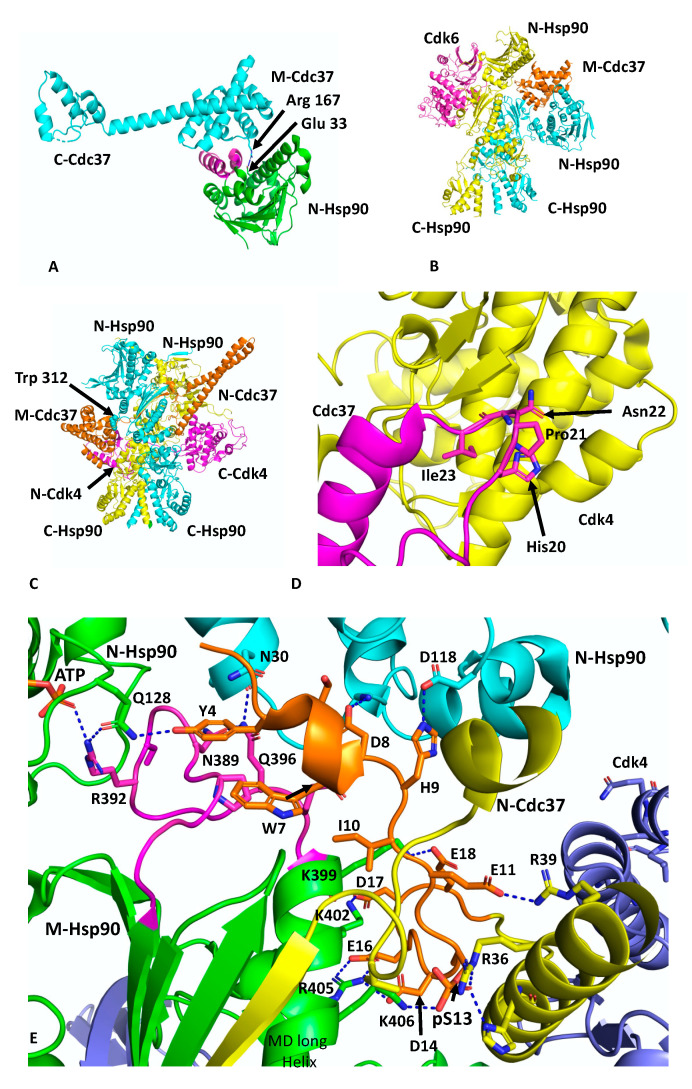
PyMol cartoons showing the interactions of Hsp90, Cdc37 and kinases. (**A**) The interaction of the MD and CTD of Cdc37 (cyan) with the NTD of Hsp90 (green). The Hsp90 lid is shown in magenta. Arg 167 of Cdc37 forms a salt bridge with Glu 33 of Hsp90. Residues are shown in stick format. (**B**) The negative stain EM structure of the Hsp90–Cdc37–Cdk4 complex, where Cdk6 was superimposed into the electron density of Cdk4 [31]. (**C**) The cryo-EM structure of the Hsp90–Cdc37–Cdk4 complex [13]. Note that the MD of Cdc37 is centered above Trp 312. (**D**) Interaction of the HPNI motif (sticks) of Cdc37 (magenta) with the C-terminal domain of Cdk4 (yellow). (**E**) Interaction of the NTD of Cdc37 (gold to yellow) with Hsp90 (green and cyan). Interactions are seen to occur with the catalytic loop (magenta) and the long helix of the MD and NTD of Hsp90. Phospho-Ser 13 (pS13) helps to stabilise the N-terminal loop of Cdc37 and forms important interactions with the base of the N-terminal end of the helix–turn–helix of the NTD of Cdc37. Some potential interactions are shown using dashed lines (blue). Blue, the CTD of Cdk4.

**Figure 3 biomolecules-12-00600-f003:**
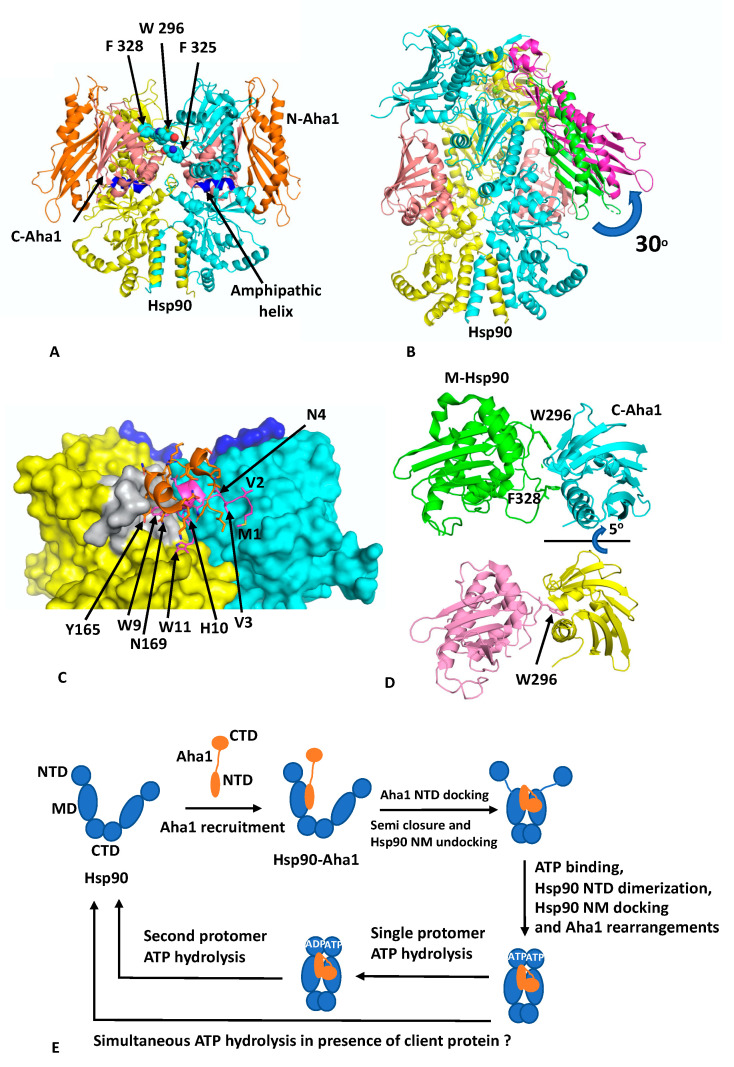
PyMol cartoons showing the Hsp90–Aha1 interactions. (**A**) apo-Hsp90 (yellow and cyan) in complex with two molecules of Aha1 (gold, N-Aha, and salmon, C-Aha1). Blue, amphipathic helix (**B**) AMPPNP-Hsp90 (yellow and cyan) in complex with two molecules of Aha1 (magenta and salmon), although only one N-terminal domain of Aha1 was observed in the complex. The rotation of the NTD of Aha1 from its apo-Hsp90 (green) to the AMPPNP-Hsp90 location (magenta) is shown. (**C**) Interaction of segments of Aha1 (magenta, residues 1–10 and gold, residues 154–170) with the NTDs of Hsp90 (yellow and cyan). Grey, surface lid segment of Hsp90. (**D**) Rotation of the CTD of Aha1 from its apo-Hsp90 (green and cyan) to the AMPPNP-Hsp90 location (salmon and yellow) results in changes of interaction from Phe 328 to Trp 296 of Aha1. (**E**) Chaperone cycles in response to Aha1 binding.

**Figure 4 biomolecules-12-00600-f004:**
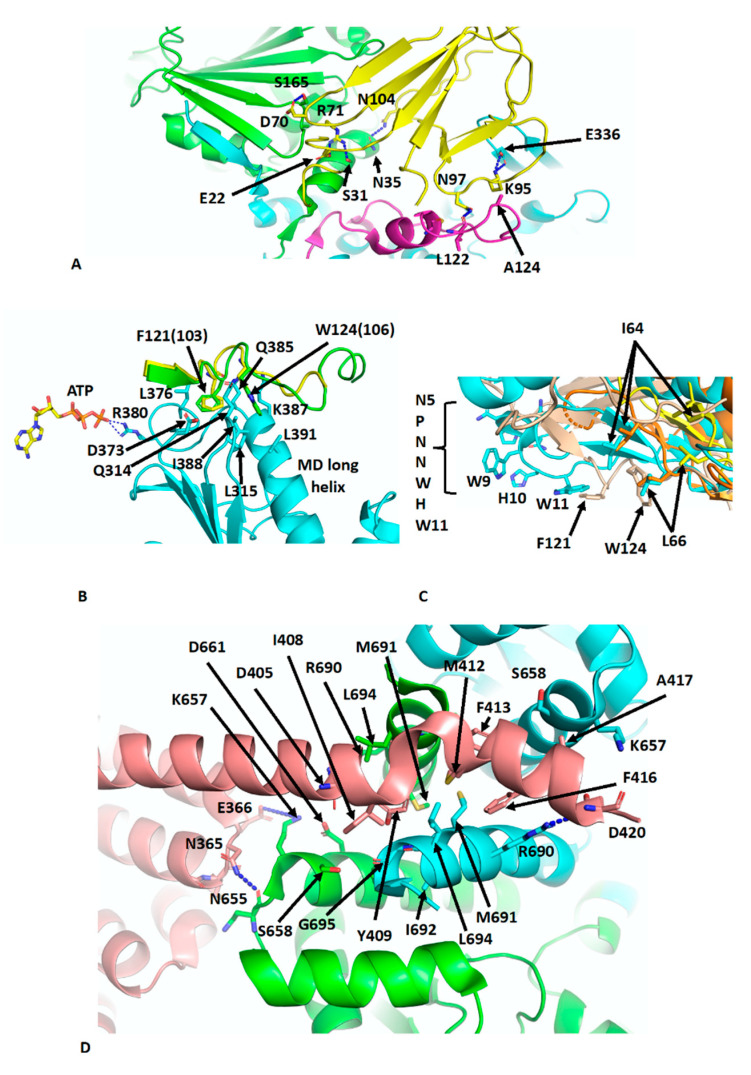
PyMol cartoons showing the Hsp90-p23 interactions. (**A**) Interaction of p23 (yellow) with the NTDs of both Hsp90 protomers (green and cyan) and the MDs of one Hsp90 protomer (magenta). (**B**) Interaction of the C-terminal unstructured segment of Sba1 (green) and p23 (yellow) with the long helix of the MD of Hsp90 (cyan). Numbers in brackets are for the human protein. (**C**) Comparison of the position of Phe 121 and W124 of Sba1 (wheat), which engage with the middle domain of Hsp90 (not shown) and the gradual engagement of specific Aha1 residues (yellow [apo-Hsp90-Aha1] to orange [Yeast Hsp90-Aha1] to cyan [Hsp90–AMPPNP–Aha1; fully tilted structure]) at similar positions with those from Sba1. Residues between Sba1 and Aha1 more or less overlap as the fully closed complex of the Hsp90–Aha1 complex is formed. Overlap of positions occur for Trp 11 and Leu 66 of Aha1, which mimic the position of Phe 121 and Trp124 of Sba1, respectively. (**D**) Interaction of the C-terminal helix 7 of FKBP51 (salmon) with the C-terminal hydrophobic pocket at the C-terminal dimer interface of Hsp90 (green and cyan). Interacting residues are shown in stick format and polar interactions as dotted blue lines.

**Figure 5 biomolecules-12-00600-f005:**
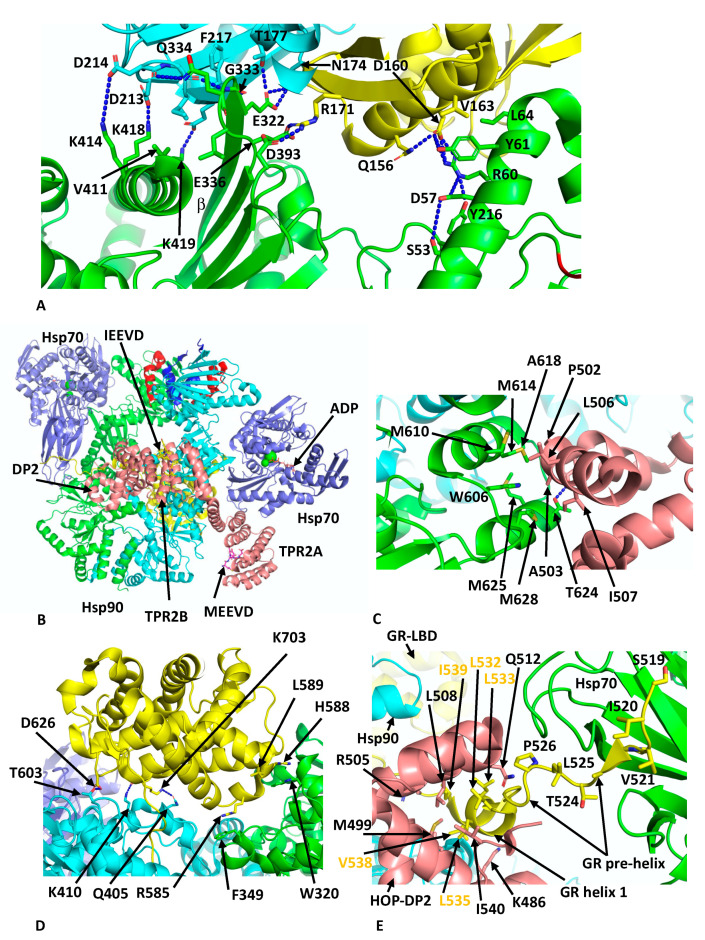
PyMol cartoons showing interactions within the Hsp90–Hsp70–HOP–GR loading complex. (**A**) Interface 1 and 2 of the Hsp90–Hsp70 interaction. A β-strand (β) from the outer face of the MD β-sheet of Hsp90 (green) inserts itself into the cleft between the subdomains IA and IIA of Hsp70 (cyan IIA and yellow, IA) in interface 1. This cleft is only available in the ADP state of Hsp70. Interface 2 is shown on the right-hand side of the panel. (**B**) The interaction of HOP and Hsp70 within the Hsp90–Hsp70–Hop-GR complex. Only three domains of Hop were visible, TPR2A, TPR2B and the DP2 domain, which appear to be fully sufficient for full GR activation. The binding of HOP is essential to maintaining the semi-closed conformation of Hsp90 and also for assembling the two bound Hsp70 molecules. Salmon, HOP; cyan and green, Hsp90 dimer; red, lid-segment of Hsp90; blue, N-terminal segments of Hsp90; and slate, Hsp70. (**C**) Interaction of the DP2 domain of HOP with the conserved client-protein binding-site residues of Hsp90. Green, Hsp90 dimer and salmon, HOP. (**D**) GR domain interactions with the Hsp90 complex. Yellow, GR; cyan and green, Hsp90 dimer; and slate, Hsp70. (**E**) Interaction of helix 1 and the pre-helix-1 segment of GR with DP2 and Hsp70, respectively. Yellow, GR; cyan, Hsp90; green, Hsp70 substrate-binding domain; and salmon, HOP DP2 domain.

**Figure 6 biomolecules-12-00600-f006:**
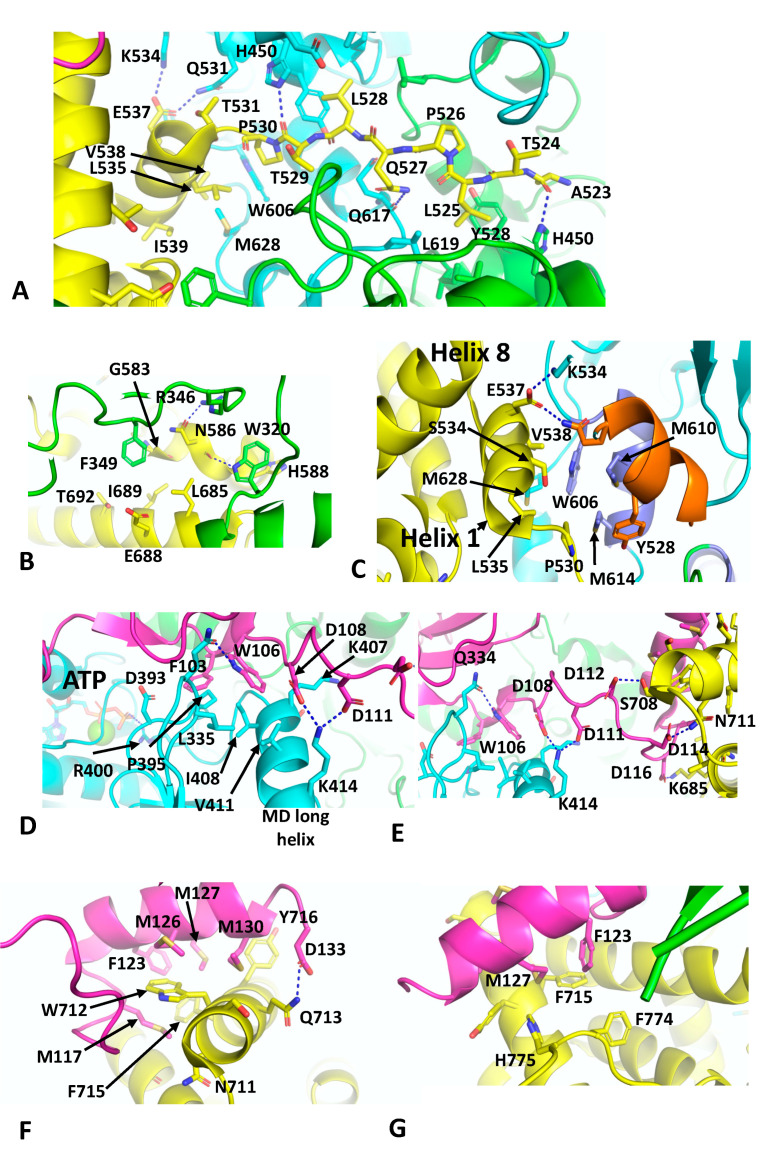
PyMol cartoon of the interactions of the Hsp90-p23-GR complex. (**A**) The pre-helix-1 held within the hydrophobic lumen of Hsp90. Yellow, GR; and cyan and green, Hsp90 dimer. (**B**) GR LBD (yellow) interacts with residues Trp 320 and Phe 349 from the adjacent Hsp90 protomer (green). (**C**) Helix 1 of GR packing against the amphipathic helix of Hsp90. Yellow, GR; cyan, Hsp90; and substrate binding sites of Hsp90 are shown in gold and slate (amphipathic helix). (**D**) The C-terminal tail of p23 (magenta) interacts with the long helix of the MD of Hsp90 (cyan and green) and the catalytic Arg 400 is in contact with the bound nucleotide (stick representation). (**E**) A relay of polar interactions between the C-terminal tail of p23 (magenta) and Hsp90 (cyan and green) and GR (yellow). (**F**,**G**) The extreme C-terminus helix of p23 (magenta) interacts with GR (yellow).

**Figure 7 biomolecules-12-00600-f007:**
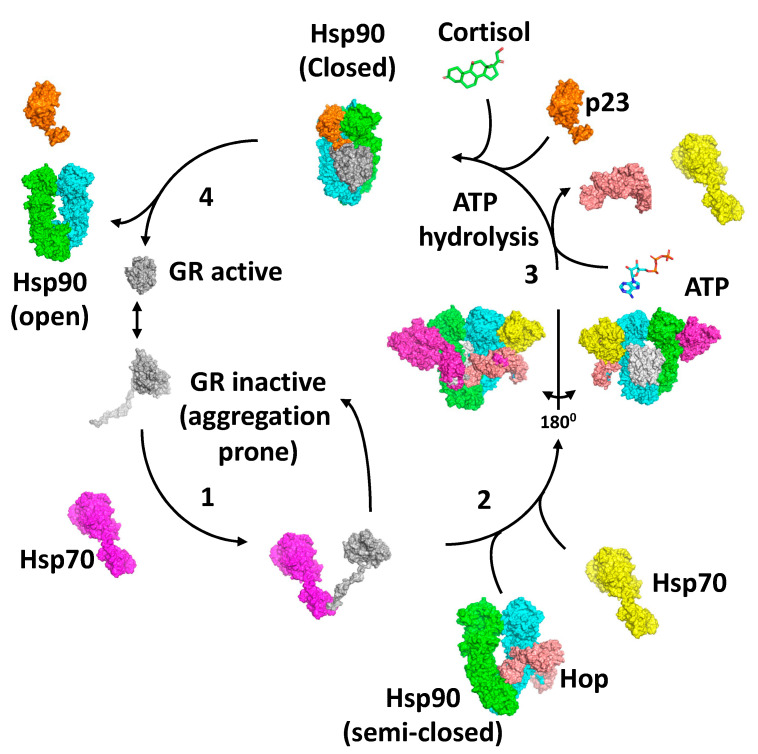
The proposed GR activation cycle. 1 to 4 represents the steps of the cycle. Grey, GR LBD; magenta and yellow, Hsp70; green and cyan, Hsp90 dimer; salmon, HOP; and orange, p23.

## Data Availability

Not applicable.

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
