# Peer review of "Advances towards Understanding the Mechanism of Action of the Hsp90 Complex"

_biomolecules, 2022, doi:10.3390/biom12050600_

Round 1

Reviewer 1 Report

In this manuscript, Dr. C.Prodromou and Dr. D.M.Bjorklund have reviewed the mechanism by which the molecular chaperone Hsp90 affects the proper folding of client proteins. The remodelling hypothesis for their activation and maturation process are analysed based on recent experimental data, most of them provided by cryo-EM studies.

The article is very well written, concepts are clearly presented and wonderfully described. Importantly, the matter of the article is vast, but the most important topics have been addressed. My only comment is related to the fact that most of the structural studies recently published in major journals using cutting-edge methodologies are confirmatory of findings already published almost 20 years ago or so by the laboratories of William Pratt and David Toft, where conventional methodologies were used. This comment is particularly valid for those studies where the stoichiometry of the TPR-Hsp90 interaction and the Hsp70-priming mechanism of GR activation are analysed. This observation does not imply that those recent studies cited here lack of importance. On the contrary, they closed a cycle of controversies in this regard. My concern is that those original quotations are missing and not properly addressed in this article. Other than this, I feel that this article is an excellent piece of work.

Author Response

I would like to thank this reviewer for their time and critical analysis of the review. I would also like to thank them for their kind words. I have now acknowledged this important historical work in reference to the recent advances. I would like to thank the reviewer for spotting this oversight. I have now added the following to the text:

The role of p23 in Hsp90 complexes was previously determined in the context of steroid hormone activation, which has been elegantly reviewed previously (Dittmar, Demady, Stancato, Krishna, & Pratt, 1997; Pratt, Morishima, Murphy, & Harrell, 2006; Pratt & Toft, 1997). The work described by these authors is not only relevant to this section, but also in the subsequent sections that look at the loading and maturation complex for glucocorticoid receptor (GR). Much of the work detailed by Pratt, WB, Toft, D, O and their co-authors showed that a Hsp90/Hsp70-based chaperone complex was responsible for regulating steroid binding, trafficking and turnover of GR. An ATP dependant activation cycle involving HOP, p23, Hsp40, FKBP51/52 the Hsp90/Hsp70 complex is able to assemble the ligand binding domain (LBD) of GR with Hsp90, in which the hydrophobic ligand-binding cleft is opened to allow access for steroid hormone binding. Much of this work has now been confirmed by the advances discussed below.

Reviewer 2 Report

The review from Prodromou and Bjorklund is well written and cover interesting aspects of a research area of great interest, therefore deserves publication in biomolecules.

I suggest carefull revision of minor issues such as:

-Sections 2 and 8, please add representative figures to better support the reader.

-Some typos should be corrected: see for example “none-the-less” or “non-the-less”, could be “nonetheless”; line 79 and 193 “form” should be “from”

Author Response

I would like to thank the reviewer for their time and critical review of the manuscript and their kind words. We have carried out a spell check and corrected the manuscript as appropriate. We have added figures as suggested. Section 2 figures have been added. However, co-ordinates for the section 8 are not apparently available and hence I was unable to address this aspect. 

Thank you for spotting my typos.

Reviewer 3 Report

Lines 73 to 165 (section 2) – This section describes the order of events leading to the acquisition of the catalytic state. Several works are cited (Schulze 2016, Ali 2006, Hessling 2009, Siligardi 2004, etc) but the picture that emerges from the description could be clearer. In a sense, the most important work (which is very nicely cited towards the end of the section) may be the effect of the R380A mutation in the Schulze 2016 study. While there is much to discover regarding the precise mechanics of the conformational chain of events, the fact that R380A abolishes all movements and events (save the ‘burst’ phase in the lid) suggests that capture of nucleotide from below by R380 occurs first and is a required for lid closure (and all subsequent events) to occur. I am curious what evidence would run counter the idea that N-M contact occurs first.

Line 75 – ‘tiggers’ should be ‘triggers’

Section beginning on Line 166 (The Hsp90-cdc37-CDK4 complex)

The subsequent section provides PDB IDs for the Hsp90-Aha1 structures that are discussed. I think it would be helpful to do the same for this section (for both the Hsp90-cdc37-CDK structures but also the structures that show the HPNV and HNVI motifs in CDK4 and BRAF (which I don’t think are cited either – although I may have missed it).

Section beginning on Line 353 (The Hsp90-p23-FKBP51 and Hsp90-Sba1 complex)

Here again, the PDB IDs are not provided the way they were for the Hsp90-Aha1 section. These would be very helpful to provide in the text.

Line 386 – ‘Lue315’ should appear ‘Leu315’.

Line 387 – the amino acid positions provided here are clarified by saying they are yeast numbers. However, these are of course different for Hsc82 and Hsp82 in this region of the protein. Hsp82 should be stated explicitly for clarity. If this ambiguity appeared elsewhere, I missed it.

Section beginning on Line 464 (The Hsp90-Hop-Hsp70-GR loading complex)

Here again, the PDB IDs are not provided the way they were for the Hsp90-Aha1 section. These would be very helpful to provide in the text.

Section beginning on Line 550 (The Hsp90-p23-GR maturation complex)

Here again, the PDB IDs are not provided the way they were for the Hsp90-Aha1 section. These would be very helpful to provide in the text (or figure legend).

Line 764 – ‘Waving’ should be spelled ‘waiving’

I really enjoyed this review. It stayed true to the stated focus and constructed a very thorough picture of the potential mechanics of the Hsp90 cycle. This will be a highly valuable resource for the field.

Author Response

I would like to thank this reviewer for their time and critical review of the manuscript. I would also like to thank them for their kind words.

1. The point about the R380 mutation is a most excellent comment. I have reviewed the structure and have had to revise my text to the following:

The association of the NM domains is critically dependent on Arg 380 (yeast Hsp90) [9]. Mutation of Arg 380 to Alanine abolishes lid closure, β-strand swap, and N/M-domain association altogether (Fig 1A) [9].Arg 380 interacts with the gamma-phosphate of the bound ATP molecule [11], but additionally forms a salt bridge with the catalytic Glu 33 (yeast Hsp90). Arg 380, therefore acts as an important interaction site linking ATP and the catalytic Glu 33 residue together. Since Glu 33 is located on helix 2, which is directly involved in the interface between the NM-domains, it would appear that Arg 380 acts as a sensor detecting the presence of bound ATP and the presence of Glu 33 in the ATP bound conformation, which subsequently allows NM association and provides stability to the NM-domain interface.

2. Tiggers has been corrected to triggers.

3. The Hsp90-Braf-Cdc37 structure including the HPNI motif are all derived form PDB 5FWM, which is now mentioned in the main text of the manuscript.

4. Other PDBs added include Hsp90-p23-GR (PDB 7KRJ), Hsp90/70-HOP-GR (PDB 7KW7) and Hsp90-p23-FKBP51 (PDB 7L7I). Other PDB have also been added as required. A PDB file for the Tau1 structure appears not to have been deposited.

5. Lue has been corrected throughout to Leu.

6. Line 387, I have clarified the situation here. The numbering was in fact for Sba1 and p23. Thank you for spotting this as it was unclear.

7. All PDB codes have been added as requested.